# A Theory of Self-Supervised Framework for Few-Shot Learning

## Abstract

Recently, self-supervised learning (SSL) algorithms have been applied to Few-shot learning(FSL). FSL aims at distilling transferable knowledge on existing classes with large-scale labeled data to cope with novel classes for which only a few labeled data are available. Due to the limited number of novel classes, the initial embedding network becomes an essential component and can largely affect the performance in practice. But almost no one analyzes why a pre-trained embedding network with self-supervised training can provide representation for downstream FSL tasks in theory. In this paper, we first summarized the supervised FSL methods and explained why SSL is suitable for FSL. Then we further analyzed the main difference between supervised training and self-supervised training on FSL and obtained the bound for the gap between self-supervised loss and supervised loss. Finally, we proposed potential ways to improve the test accuracy under the setting of self-supervised FSL.

## 1 Introduction

Recently, the self-supervised learning (SSL) algorithms have been applied to the FSL. The purpose of FSL is to extract transferable knowledge from existing classes with large-scale label data to deal with novel classes with only a few labeled data. The initial embedded network becomes an essential component and will greatly affect performance because of the limited number of novel classes. In practice, SSL greatly enhances the generalization of FSL method and increases the potential for industrial application. Once combining SSL and FSL, we only need to collect a large amount of related unlabeled data and a few data on the new task to obtain a model with good generalization performance on the new task. In theory, it is difficult to analyze the performance of self-supervised pre-trained models on multiple downstream tasks. Because the downstream task itself involves a large amount of data with different data distribution from the primary training data distribution, such as multi-view SSL. Besides, downstream tasks and self-supervised tasks may be quite different, such as classification and segmentation, which further increases the difficulty of theoretical analysis. However, when back to the purpose of SSL, which is to learn a good pretrain model that can be transferred to different tasks, we find FSL also focus the same purpose to get an initialization model that can achieve good results with a few data on a new task by a simple classifier (such as a mean classifier). Thus, FSL tasks is suitable for evaluation of the effect of SSL.

The main research direction about when and why self-supervised methods improves FSL is to compare the performance of different self-supervised methods through experiments. Almost no one analyzes why a pre-trained embedded network with self-supervised training can provide a representation for downstream FSL tasks in theory. We believe that theoretical analysis is necessary. For example, MoCo uses momentum update to greatly expand the size of the key-value dictionary, thereby improving the effect. But we don't know why the key-value dictionary needs to be large enough. Is the batch size really the bigger the better? SimCLR proposes a head layer to calculate the contrastive loss, instead of directly comparing the representations. Why is this method effective? We find that although self-supervised learning researchers have made great progress, analysis about why SSL works is halted at experimental and empirical conclusions due to the lack of theorical analysis. Therefore, we think it is necessary and useful to analyze self-supervised learning theoretically.

We analyze the self-supervised training process via the specific application scenario of FSL. Under this settings, we avoid the complexity of downstream tasks, and can directly judge the quality of self-supervised learning by the performance of new few-shot tasks. Our main intuition is to quantify the gap between self-supervised learning and supervised training on FSL tasks by constructing supervised metrics corresponding to self-supervised tasks. We find that the self-supervised training loss is actually an upper bound of the supervised metric loss function (Theorem 1). It means that if we can reduce the self-supervision loss small enough, we can control the model's supervision loss on the training data. And because FSL methods has good generalization on similar downstream tasks, we conclude that self-supervised training can also have good generalization on similar tasks, even if the categories of training tasks and test tasks are different.

Unfortunately, it is often difficult to minimize the training loss of self-supervision. Contrastive-based SSL method samples different augment data as query and positive data, and others as negative data. Those false negative data have the same class as query. This part of training loss introduced by the false negative data limits our performance. We separate the negative samples in the self-supervised training into true negative samples and false negative samples. For true negative samples, we assume that loss can be small enough by suitable models and optimizers. As for false negative samples, we bound this loss by the intra-class deviation. This part is also the difference between self-supervised learning and supervised learning (Theorem 2). We should control the intra-class deviation of these false negative samples while training according to Theorem 2.

Finally, we discuss potential ways to improve test accuracy under self-supervised FSL settings. First, we suggest that the larger the batch size is, the better, but within a certain range. Second, increasing the number of support samples is beneficial to reducing the within-class variance of false negative samples for good test performance. Technically, we set the different augmented data as the support samples from the same class. Third, we need to choose unsupervised training data whose number of categories are large, because large categories will reduce the probability of us sampling false negative samples. We also introduce the limitations of our theory.

Ideally, one would like to know whether a simple contrastive self-supervised framwork can give representations that competable with those learned by supervised methods. We show that under the two assumptions, one can get a test performance close to supervised training. Experiments on Omniglot also support our theoretical analysis. For instance, the self-supervised framework reaches the accuracy of $98.23\%$ for 5-way 5-shot classification on Omniglot, which is quite competitive compared to $98.83\%$ achieved by supervised MAML.

## 2 PRELIMINARY KNOWLEDGE AND ASSUMPTION

### 2.1 SUMMARY OF SUPERVISED FSL METHODS

In the typical few-shot scenario introduced by Vinyals et al. (2016), the model is presented with episodes composed of a support set and a query set. The support set contains concepts about the categories into which we want to classify the queries. In fact, models are usually given five categories (5-way), and one (one-shot) or five (five-shot) images per category. During training, the model is fed with these episodes and it has to learn to correctly label the query set given the support set. The category sets seen during training, validation, and testing, are all disjoint. This way we know for sure that the model is learning to adapt to any data and not just memorizing samples from the training set. Although most algorithms use episodes, different algorithm families differ in how to use these episodes to train the model. Recently, transfer learning approaches have become the new state-of-the-art for few-shot classifications. Methods like Gidaris & Komodakis (2018), pre-train a feature extractor and linear classifier in a first stage, and remove its last FC layer, then fix the feature extractor and train a new linear classifier on new samples in the fine-tuning stage. Due to its success and simplicity, transfer learning approaches have been named "Baseline" on two recent papers Chen et al. (2019); Dhillon et al. (2019).

We mark the feature extractor in the first stage as $f_q$, and the linear classifier of Chen et al. (2019) on new samples in the fine-tuning stage as $y = f_q(x)^{\mathrm{T}}\mathbf{W}, \mathbf{W} = [\mathrm{w}_1, \mathrm{w}_2, \ldots, \mathrm{w}_c] \in \mathbb{R}^{d \times c}$. The classifier in Chen et al. (2020b) is a mean classifier with the weight as the centroid of features

from the same class. Let $S_c$ denote the few-shot support samples in class $c$, then they have $w_c = \frac{1}{|S_c|} \sum_{x \in S_c} f_q(x)$. In this paper, we take a generalized mean classifier with $w_c = \frac{1}{|S_c|} \sum_{x \in S_c} f_k(x)$. Because of the arbitrariness and complexity of $f_k$, this generalized mean classifier is nearly an arbitrary linear classifier.

## 2.2 Assumptions in self-supervised FSL

**Assumption 1** *Mean classifier is good enough for evaluation.*

We usually analyze the effectiveness of self-supervised with the results on downstream tasks Liu et al. (2020); Jing & Tian (2020). But another different downstream task usually makes the theoretical analysis more difficult. And a complex classifier usually performs better than mean classifier on most of tasks Liu et al. (2020); Jing & Tian (2020), i.e., the classification on ImageNet. Therefore, we consider a simple way to analyze self-supervised learning. We assume that only the mean classifier is used for classification during the test phase. When the $f_k$ in mean classifier is complex enough, it is equivalent to linear classifier which used in Baseline++. Even if $f_k$ is simply the same as $f_q$, this mean classifier is consistent with the classifier in metric-based FSL methods, such as ProtoNets Snell et al. (2017). The mean classifier is suitable for self-supervised FSL because of the the good performance, simple form, convenient measurement and wide usability.

We assume that the mean classifier performs well enough during testing if decreasing supervised training loss enough. Our hypothesis has been verified on Baseline++ Chen et al. (2019), ProtoNet Snell et al. (2017) and some other transfer-based FSL methods Dhillon et al. (2019); Li et al. (2019a); Hu et al. (2020). Our paper mainly focuses on analyzing the difference between supervised training loss and self-supervised training loss on the feature extractor $f_q$ and the mean classifier based on $f_k$, regardless of how to generalize from the training set to the test set. We just make an assumption that generalization remains to work based on existing supervised FSL methods.

**Assumption 2** *The training data is balanced in different classes.*

The main work of this paper is to analyze self-supervised loss. We assume that each training sample is drawn from a certain data distribution. We regard the sampling process as two steps, first randomly sample the categories, and then select samples from these categories for training. In order to analyze the difference between the self-supervised loss and the supervision loss, we mark these samples with their labels in supervised dataset, and then selecte one sample for each class to construct a dynamic $N$-way 1-shot supervised training task. When assumption 2 is satisfied, that is, the number of samples in each class is the same, our theoretical results are relatively simple and clear.

In practice, the assumption is also acceptable because we can collect data by specifying some keywords to ensure that the data is roughly balanced. Please note that whether this assumption is necessary is still worth further analysis. There may be different annotations in different self supervised pretext tasks. Especially in the FSL, the categories in the training set and the test set do not overlap, so the assumption of balanced data is more likely to be removed. This paper assume this assumption is true and do not further analysis the necessity.

The above two assumptions are the basis of our subsequent analysis. Our theoretical analysis is also applicable to other scenarios that satisfied the two assumptions, not just FSL. We chose self-supervised FSL in our experiment because these two assumptions have been adopted in FSL.

## 3 Theories for self-supervised loss and supervised loss

### 3.1 Contrastive self-supervised training framework

Given an unlabeled dataset $\mathcal{A}$, self-supervised training method, like MoCo He et al. (2019) creates many synthetic query-key matching tasks on-the-fly by randomly sampling $N_K$ data at a time and then augmenting them. A basic consideration is that two synthetic data, $x^+ = \text{Aug}(x)$ and $x^q = \text{Aug}'(x)$, who are augmented from the same ancestor $x$, hold the same class label. In this case, one of the $N_K$ data is randomly selected to be the positive data, and its two augmented data are taken as the query and the positive samples, respectively, while the synthetic data augmented from the remainder $N_K - 1$ data are treated as negative samples. After that, the query encoder $f_q$ maps the

query into $q = f_q(x^q)$, and the key encoder $f_k$ maps the positive sample into $k^+ = f_k(x^+)$, and the negative samples into $k_i^- = f_k(x_i^-)$, $i = 1, \ldots, N_K - 1$. These output vectors $q, k^+, k_i^-$ are normalized by its L2-norm, followed by a metric loss:

$$\mathcal{L} = \log(1 + \sum_{i=1}^{N_K-1} \exp(\mu q^{\mathrm{T}} k_i^- - \mu q^{\mathrm{T}} k^+)), \tag{1}$$

where $\mu$ is a learnable metric scaling scalar Oreshkin et al. (2018) in the hope of facilitating metric training. As a note, in our experiments, the metric loss $\mathcal{L}$ is evaluated based on multiple query data in a mini-batch manner. Please refer to Appendix C for more details.

In our paper, we assume that the queries from the same class have the same distribution, and the keys from the same class have the same distribution. The query and keys depend on the specific self-supervised pretext task. The input $x_q$ and $x_k$ can be images Hadsell et al. (2006); Wu et al. (2018); Ye et al. (2019), or context consisting a set of patches Oord et al. (2018) [44]. The networks $f_q$ and $f_k$ can be identical Hadsell et al. (2006); Ye et al. (2019); Chen et al. (2020a), partially shared Oord et al. (2018); Hjelm et al. (2018); Rezaabad & Vishwanath (2019), or different Tian et al. (2019).

## 3.2 SUPERVISED AND SELF-SUPERVISED TRAINING LOSS.

Assume the dataset $\mathcal{A}$ to be supervised, the same framework can be trained in a supervised few-shot learning manner using this $\mathcal{A}$. We will show that the self-supervised loss is an upper bound for supervised evaluation metric, and prove that minimizing unsupervised loss makes sense in Section 4.

**Self-Supervised Metric (SSM) for Representations** SSM accesses to a flow of unsupervised tasks $\mathcal{T}_t$ which contain augumented data $\{x^q, x^+, x_1^-, \ldots, x_{N_K-1}^-\}$. We mark their ground-truth labels by $\mathcal{C}_{\mathrm{U}} = \{c^q, c^+, c_1^-, \ldots, c_{N_K-1}^-\}$, respectively. Note that $x^q$ and $x^+$ are drawn from the same data distribution $\mathcal{D}_{c^+}$ (since $c^q = c^+$) while negative $x_i^-$ are from $\mathcal{D}_{c_i^-}$. Let $I = \{1, \ldots, N_K - 1\}$ be the set of indices of negative data, the unsupervised loss in Eq. (1) can be rewritten as

$$\mathcal{L}_{\mathrm{U}} = \mathbb{E}_{q,k^+,k_i^-} \log(1 + \sum_{i \in I} \exp(\mu q^{\mathrm{T}} k_i^- - \mu q^{\mathrm{T}} k^+)). \tag{2}$$

**Supervised Metric for Representations.** The quality of the representation function $f_q, f_k$ is evaluated by its performance on a multi-class classification task using *linear classification*. Let $\mathcal{C}$ denote the set of class label with prior distribution $\rho$. Assume that the augumented data $x \in \mathcal{X}$ is drawn from the data distribution $\mathcal{D}_c$, where $c \sim \rho$. Now considering one $N$-way task $\mathcal{T}_{sup}$ on $N$ different classes $\mathcal{C}_{\mathrm{sup}} = \{c_1, \ldots, c_N\}$, its multi-class classifier is denoted as the funtion $g : \mathcal{X} \to \mathbb{R}^N$. The softmax-based cross-entropy loss on data pair $(x, y)$ can be rewritten as

$$\mathcal{L}_{\mathrm{sup}}(g, x, y) = \log(1 + \sum_{y' \neq y} \exp(g(x)_{y'} - g(x)_y)), \tag{3}$$

where $g(x)_y$ is the $y$-th element of the vector $g(x)$. It is a general intuition that the data pair $(x, y)$ with right label $y$ has greater confidence than the data pair $(x, y')$ with wrong label $y'$.

We choose the linear classifier as *mean classifier* $g(x)_c = \mu q^{\mathrm{T}} p_c$, where $\mu$ is a scaling scalar and $q = f_q(x)$ is the query representation, and $p_c = \mathbb{E}_{x \sim \mathcal{D}_c}[f_k(x)]$ is the mean of representation of inputs with label $c$. That is, $f_q$ is for the feature extraction, and $p_c$ is the weight of linear classifier. Then, the expected supervised metric loss in terms of $f_q, f_k$ on $N$-way tasks is

$$\mathcal{L}_{\mathrm{sup}} = \mathbb{E}_{c \sim \rho, x \sim \mathcal{D}_c} \log(1 + \sum_{c' \neq c} \exp(\mu q^{\mathrm{T}} p_{c'} - \mu q^{\mathrm{T}} p_c)). \tag{4}$$

Our goal is to find a unsupervised training method to decrease the value of the evaluation metric $\mathcal{L}_{\mathrm{sup}}$. Previous supervised FSL methods deal with the evaluation metric in different ways. ProtoNets conducts an episode training by using the evaluation metric as supervised loss function but replacing scaled dot products with Euclidean distance. Chen Chen et al. (2019) shows that few-shot algorithms, such as MatchNets, ProtoNets, RelationNets, and MAML are based on episode training on the variants of $\mathcal{L}_{\mathrm{sup}}$. Recent researchs Dhillon et al. (2019); Chen et al. (2019; 2020b) have

found the basic episode training on $\mathcal{L}_{\text{sup}}$ with proper pretrain outperforms most of methods. The core of FSL is to generate a flow of supervised $N$-way $M$-shot tasks, and training with viriants of $\mathcal{L}_{\text{sup}}$ on each task. For the generality and flexibility of our theories, we will prove that contrastive self-supervised training methods with loss function Eq. 2 is essentially reducing $\mathcal{L}_{\text{sup}}$ on training data. We analyze the evaluation metric with a class-wise prototype $p_c$ rather than a episodic mean of support samples. It is proved to be a better choice of loss function due to explicitly reducing intra-class variations Chen et al. (2019).

## 4 GAPS BETWEEN SUPERVISED AND SELF-SUPERVISED TRAINING LOSS

It can be seen that Eq. (2) has a similar form with Eq. (4). As a first step, we show that $\mathcal{L}_{\text{U}}$ bounds the $\mathcal{L}_{\text{sup}}$ in an ideal situation. This conclusion indicates that it makes sense to minimize the unsupervised SSM loss $\mathcal{L}_{\text{U}}$. We represent the upper bound for the supervised loss $\mathcal{L}_{\text{sup}}$ by Theorem 1.

**Theorem 1** $\forall f_q, f_k \in \mathcal{F}$,
$$\mathcal{L}_{\text{sup}} \leq \gamma_0 \mathcal{L}_{\text{U}} + \delta, \tag{5}$$
*where $\gamma_0$, $\delta$ are constants depending on the class distribution $\rho$. When $\rho$ is uniform and $|\mathcal{C}| \to \infty$, then $\gamma_0 \to 1$, $\delta \to 0$.*

*Proof.* The key point for proof is the use of Jensen's inequality since $\ell(\boldsymbol{v}) = \log(1 + \sum_i \exp(\boldsymbol{v}_i)), \forall \boldsymbol{v} \in \mathbb{R}^{N_K - 1}$ is a convex function ($\boldsymbol{v}_i$ is the $i$-th element of $\boldsymbol{v}$), that is,

$$
\begin{aligned}
\mathcal{L}_{\text{U}} &= \underset{q}{\mathbb{E}} \underset{k^+, k_i^-}{\mathbb{E}} \log(1 + \textstyle\sum_{i \in I} \exp(\mu q^{\text{T}} k_i^- - \mu q^{\text{T}} k^+)) \\
&\geq \underset{q, c^+, c_i^-}{\mathbb{E}} \log(1 + \textstyle\sum_{i \in I} \exp(\mu q^{\text{T}} p_{c_i^-} - \mu q^{\text{T}} p_{c^+})).
\end{aligned} \tag{6}
$$

However, unsupervised data label $\mathcal{C}_{\text{U}}$ may contain duplicate classes and even false negative classes. Clearly, we divide $I$ into two disjoint subsets, true negative index set $I^- = \{i \in I | c_i^- \neq c^+\}$ and false negative index set $I^+ = \{i \in I | c_i^- = c^+\}$. We define $\mathcal{C}_{\text{uni}}$ as the label set after de-duplicating class labels in $\mathcal{C}_{\text{U}}$, $\mathcal{C}_{\text{uni}} \subseteq \mathcal{C}_{\text{U}}$. Since $\ell(\{\boldsymbol{v}_i\}_{i \in I_1 \cup I_2}) := \log(1 + \sum_{i \in I_1 \cup I_2} \exp(\boldsymbol{v}_i)) \geq \ell(\{\boldsymbol{v}_i\}_{i \in I_1})$, $\forall I_1, I_2 \subseteq I$, we could decompose Eq. (6) into

$$
\begin{aligned}
&\underset{q, c^+, c_i^-}{\mathbb{E}} \log(1 + \textstyle\sum_{i \in I} \exp(\mu q^{\text{T}} p_{c_i^-} - \mu q^{\text{T}} p_{c^+})) \\
&\geq P(I^+ = \emptyset) \underset{q, c^+}{\mathbb{E}} \left[ \ell(\{\mu q^{\text{T}} p_c - \mu q^{\text{T}} p_{c^+}\}_{c \in \mathcal{C}_{\text{uni}} \backslash c^+}) | I^+ = \emptyset \right] \\
&\quad + P(I^+ \neq \emptyset) \underset{c^+}{\mathbb{E}} [\log(1 + |I^+|) | I^+ \neq \emptyset].
\end{aligned} \tag{7}
$$

The first expectation in Eq. (7) is actually the supervised loss $\mathcal{L}_{\text{sup}}$ in Eq. (4) by regarding $\mathcal{C}_{\text{sup}} := \mathcal{C}_{\text{uni}}$. Combining this result with Eq. (6) (7), we obtain the inequality in Theorem 1 with $\gamma_0 = \frac{1}{P(I^+ = \emptyset)}$, $\delta = -\frac{P(I^+ \neq \emptyset)}{P(I^+ = \emptyset)} \underset{c^+}{\mathbb{E}} [\log(1 + |I^+|) | I^+ \neq \emptyset]$. When $\rho$ is uniform and $|\mathcal{C}| \to \infty$, then $P(I^+ \neq \emptyset) \to 0$. Please refer to Appendix B.1 for more details.

The supervised loss is built from unsupervised training. We consider the following tweak in the way: sample the class set $\{c^+, c_1^-, \ldots, c_{N_K - 1}^-\}$ from the class distribution $\rho$, and random select one class as positive class, and consider the episode that all negative classes are different from the positive class ($I^+ = \emptyset$), then sample one data from one negative class independently to get $|\mathcal{C}_{\text{sup}}|$-way 1-shot supervised loss. Notice that this loss is about separating $c^+$ from the total class set, we calculate the probability by symmetrization under the assumption that training dataset are balanced. We derive the above encouraging result about the relationship between SSM loss and supervised loss. Thus, we can decrease the supervised loss by minimizing the unsupervised SSM loss $\mathcal{L}_{\text{U}}$. However, $\mathcal{L}_{\text{U}}$ can not be small enough if false negative keys present in some tasks and the proportion of false negative data is considerable, especially for the task scenarios whose label space $|\mathcal{C}|$ is small (*e.g.*, *mini*ImageNet). In that case, minimizing $\mathcal{L}_{\text{U}}$ will meet a theoretical bottleneck since the loss on the false negative data can not be decreased enough.

To explore the effect of some inputs of the same class as positive data sneaking into the negative data, we further decompose the SSM loss $\mathcal{L}_{\text{U}}$ into two terms: (1) $\mathcal{L}_{\text{U}}^-$, the loss on all true negative data $x_i^-$, $i \in I^-$. These data has the different labels from the positive data. (2) $\mathcal{L}_{\text{U}}^+$, the loss on all

false negative data $x_i^-, i \in I^+$. These data have the same label as the positive data. Theorem 2 does reveal the underlying factors behind the explicit gap between $\mathcal{L}_{\text{sup}}$ and $\mathcal{L}_{\text{U}}$. We define a notation of intra-class deviation as $s(f_k) := |\mu|\mathbb{E}_c[\mathbb{E}_{x \in \mathcal{D}_c}\|f_k(x) - p_c\|_2^2]^{1/2}$, and show that $s(f_k)$ can bound $\mathcal{L}_{\text{U}}^+$. Then, we get a new narrow bound by Theorem 2.

**Theorem 2** $\forall f_q, f_k \in \mathcal{F}$,
$$\mathcal{L}_{\text{sup}} \leq \gamma_0 \mathcal{L}_U^- + \gamma_1 s(f_k), \tag{8}$$
*where $\gamma_0$, $\gamma_1$ are constants depending on the class distribution $\rho$. When $\rho$ is uniform and $|\mathcal{C}| \to \infty$, then $\gamma_0 \to 1$, $\gamma_1 \to 0$.*

*Proof.* The key point for proof is that $\ell(\{v_i\}_{i \in I_1 \cup I_2}) \leq \ell(\{v_i\}_{i \in I_1}) + \ell(\{v_i\}_{i \in I_2}), \forall I_1, I_2 \subseteq I$,
$$\begin{aligned}
\mathcal{L}_{\text{U}} &\leq \mathop{\mathbb{E}}_{q,k^+,k_i^-} \log(1 + \textstyle\sum_{i \in I^-} \exp(\mu q^{\text{T}} k_i^- - \mu q^{\text{T}} k^+)) \\
&+ \mathop{\mathbb{E}}_{q,k^+,k_i^-} \log(1 + \textstyle\sum_{i \in I^+} \exp(\mu q^{\text{T}} k_i^- - \mu q^{\text{T}} k^+)) \\
&:= \mathcal{L}_{\text{U}}^- + \mathcal{L}_{\text{U}}^+,
\end{aligned} \tag{9}$$

where the first expectation is $\mathcal{L}_{\text{U}}^-$ and the second one is $\mathcal{L}_{\text{U}}^+$ (imagining that true negative data and false negative data have been separated during training). With the following inequalities $\ell(\{v_i\}_{i \in I_1}) \leq \log(1 + |I_1|) + \max\{\max\{v_i\}_{i \in I_1}, 0\}$, and $\max\{v_i\}_{i \in I_1} \leq |\max\{v_i\}_{i \in I_1}| \leq \max\{|v_i|\}_{i \in I_1} \leq \sum_{i \in I_1} |v_i|$, we can get the following inequality:
$$\begin{aligned}
\mathcal{L}_{\text{U}}^+ &\leq \mathop{\mathbb{E}}_{q,k^+,k_i^-} \left[ \log(1 + |I^+|) + \textstyle\sum_{i \in I^+}(|\mu q^{\text{T}} k_i^- - \mu q^{\text{T}} k^+|) \right] \\
&= P(I^+ \neq \emptyset)\mathop{\mathbb{E}}_{c^+} \left[ \log(1 + |I^+|)|I^+ \neq \emptyset \right] \\
&+ \mathop{\mathbb{E}}_{q,k^+,k_i^-} \left[ \textstyle\sum_{i \in I^+} |\mu q^{\text{T}} k_i^- - \mu q^{\text{T}} k^+| \right].
\end{aligned} \tag{10}$$

By combining Eq. (5) with Eq. (9) and Eq. (10), we get
$$\begin{aligned}
\mathcal{L}_{\text{sup}} &\leq \gamma_0(\mathcal{L}_{\text{U}}^- + \mathcal{L}_{\text{U}}^+) + \delta \\
&\leq \gamma_0 \mathcal{L}_{\text{U}}^- + \gamma_0 \mathop{\mathbb{E}}_{q,k^+,k_i^-} \left[ \textstyle\sum_{i \in I^+} |\mu q^{\text{T}} k_i^- - \mu q^{\text{T}} k^+| \right] \\
&= \gamma_0 \mathcal{L}_{\text{U}}^- + \gamma_0 \mathop{\mathbb{E}}_{c^+} \mathop{\mathbb{E}}_{q,k^+,k_i^- \sim c^+} \left[ \textstyle\sum_{i \in I^+} |\mu q^{\text{T}} k_i^- - \mu q^{\text{T}} k^+| \right].
\end{aligned} \tag{11}$$

When the class distribution $\rho$ is uniform, we have $\mathbb{E}|I^+| = (N_K - 1)/|\mathcal{C}|$ for any class. Considering that the representations are normalized to satisfy $\|q\|_2 = 1$, thus the right expectation in Eq. (11) can be bound by $s(f_k)$, that is,
$$\mathop{\mathbb{E}}_{c^+} \mathop{\mathbb{E}}_{q,k^+,k_i^- \sim c^+} [\sum_{i \in I^+} |\mu q^{\text{T}} k_i^- - \mu q^{\text{T}} k^+|] \leq \sqrt{2}\mathbb{E}|I^+| \cdot s(f_k). \tag{12}$$

Combining Eq. (11) and Eq. (12), Theorem 2 can be proved, where $\gamma_1 = \frac{\sqrt{2}\gamma_0(N_K-1)}{|\mathcal{C}|}$, $\gamma_0 = \frac{1}{P(I^+=\emptyset)}$. When $\rho$ is uniform and $|\mathcal{C}| \to \infty$, then $P(I^+ \neq \emptyset) \to 0$, $\gamma_0 \to 1$, $\gamma_1 \to 0$. Please refer to Appendix B.2 for more details.

Compared to Theorem 1, Theorem 2 indicates that the supervised loss $\mathcal{L}_{\text{sup}}$ is bounded by two explicit parts. The first one is $\mathcal{L}_{\text{U}}^-$, which measures the similarity between the query data with the positive data and true negative data. It is somehow like dynamic $N$-way $M$-shot supervised training. The difference is that the value of $N$ and $M$ might change along with the number of true negative data. The second is $s(f_k)$, which acts as the penalty for representation ability by measuring the intra-class representation deviation. Moreover, the $\gamma_1 s(f_k)$ is an explicit gap distancing unsupervised SSM loss from supervised loss. Theoretically, if given an unsupervised set with infinite classes and data, the performance achieved by SSM can be very close to that by supervised training.

## 5 DISCUSSION ON SELF-SUPERVISED FSL

**Impact of the value $N$, $M$ in dynamic $N$-way $M$-shot.** In Section 4, we regard $\mathcal{L}_{\text{U}}^-$ as dynamic $N$-way $M$-shot supervised training. It depends on the sampled self-supervised true negative training

| $N_K$ | 128 | 512 | 2048 | 8192 |
|---|---|---|---|---|
| Omniglot | 90.587±0.006 | 91.475±0.004 | **92.013 ± 0.005** | 92.001±0.008 |
| *mini*ImageNet | 38.964±0.012 | **41.783 ± 0.007** | 40.275±0.008 | 39.654±0.007 |

Table 1: Accuracy (%) averaged over 1000 random 5-way 1-shot test tasks.

| $M$ | 1 | 2 | 3 | 4 |
|---|---|---|---|---|
| Omniglot | 98.091±0.007 | 98.163±0.005 | 98.234 ±0.006 | 98.258±0.005 |
| *mini*ImageNet | 59.128±0.009 | 60.016±0.008 | 60.118±0.006 | 60.121±0.007 |

Table 2: Accuracy (%) averaged over 1000 random 5-way 5-shot test tasks.

data. In supervised FSL methods, the performance increases as the $N$ or $M$ increases. So how to increase $N$ and $M$ in self supervised training, and what is the improvement? We can increase $N$ by increasing the total negative samples $N_K$. But $\gamma_1$, the coefficient of the gap, also increases. Thus we suggest that in the self-supervised FSL, we do not need to use a particularly large $N_K$. Experiments in Table 1 on Omniglot and *mini*ImageNet show that the best $N_K$ is 2048, 512, respectively.

Most of self-supervised methods take one positive key for one query while FSL usually set $M$ support samples as positive keys. ProtoCLR Medina et al. (2020) proposed that original images x serve as class prototypes around which their $Q$ augmentations should cluster. And $Q = 3$ showed the best performance in their experiments. Supposed that we replace each key representations with the average of $M$ different key representations from different augmentations of the same input $x$. The new intra-class deviation $s(\bar{f}_k) := s(f_k)/\sqrt{M}$ reduces the gap between self-supervised and supervised training.

**Impact of the class number** $|\mathcal{C}|$. The class number affects the coefficient of gap. Assuming that the class number is infinite, then the probability of sampling false negative data is zero, thus we have $\gamma_0 \to 1, \gamma_1 \to 0$. Concretely, the unsupervised $\mathcal{A}$ of *mini*ImageNet only contains 64 classes while that of Omniglot involves up to 4800 classes, which leads to a smaller $\gamma_1$ for Omniglot (larger $|\mathcal{C}|$ implies larger $P(I^+ = \emptyset)$, larger $P(I^+ = \emptyset)$ implies smaller $\gamma_0$, larger $|\mathcal{C}|$ + smaller $\gamma_0$ imply smaller $\gamma_1$). Assume that there is no significant difference about the representation ability of the backbone on two datasets (even if the intra-class deviation on Omniglot is intuitively smaller than that on *mini*ImageNet since Omniglot is more easy), the gap $\gamma_1 s(f_k)$ on Omniglot is smaller than that on *mini*ImageNet. Because the more categories, the more information introduced, it is difficult to design a fair comparative experiment. Still, we believe self-supervised methods is suitable for large number of categories according to the theory.

**Impact of the imbalance data.** We assume training data is balanced to get simple and clear coefficient $\gamma_0, \gamma_1$ in the gap. From Eq. 12, the intra-class deviations of different classes are weighted by the expected ratio of false negative samples in all negative samples. Moreover, The generalization performance may also be affected by unbalanced data training. Because the weight of each sample in the supervision loss varies in different categories.

**Limitations of the theories.** (1) For simplicity, we only consider the mean classifier, and the mean classifier in the supervised framework may become a bottleneck. Not all FSL methods take the mean classifier or linear classifier and training the model in two stages. We do not discuss the relationship between different FSL methods and the supervision loss in our paper, which may limit the theory generalizing to other FSL methods. (2) We have not carried out abundant experiments to verify the theory. There are still many obstacles in guiding experiments with theory. We think that collecting large categories of unsupervised data, increasing the number of negative samples in a certain extent, and using multiple augmentations to replace the original keys, will help to reduce the gap between supervision and self-supervised training. (3) The supervised loss is actually an evaluation metric, slightly different from the training loss used in FSL methods. Therefore, the convergence and generalization of the framework need to be further verified.

## 6 RELATED WORK

**Few-Shot Learning** Compared to the common machine learning paradigm that involves large-scale labeled training data, the development of FSL is tardy due to its intrinsic difficulty. Early works for FSL were based on generative models that sought to build a Bayesian probabilistic framework Fei-Fei et al. (2006); Lake et al. (2015). Recently, more works were from the view of meta-learning, which can be roughly summarized into five sub-categories: *learn-to-measure* (*e.g.*, MatchNets Vinyals et al. (2016), ProtoNets Snell et al. (2017), RelationNets Yang et al. (2018)), *learn-to-finetune* (*e.g.*, Meta-Learner LSTM Ravi & Larochelle (2017), MAML Finn et al. (2017)), *learn-to-remember* (*e.g.*, MANN Santoro et al. (2016), SNAIL Mishra et al. (2018)), *learn-to-adjust* (*e.g.*, MetaNets Munkhdalai & Yu (2017), CSN Munkhdalai et al. (2018)) and *learn-to-parameterize* (*e.g.*, DynamicNets Gidaris & Komodakis (2018), Acts2Params Qiao et al. (2018)). These methods consistently need to create meta-training tasks on a large-scale supervised auxiliary set to obtain an FSL model that can be transferred across tasks.

**Contrastive self-surpervised representation learning** Our analysis is based on contrastive representation learning. It is a classic machine learning topic Barlow (1989) that aims to acquire a pretrained representation space from unsupervised data and works as a pre-bedding for downstream supervised learning tasks. It can be traced back to Hadsell et al. (2006), these methods learn representation by comparing positive and negative pairs. Wu et al. (2018) suggests using a repository to store instance class representation vectors, which is a method adopted and extended in recent papers. Other studies have explored the use of intra batch samples instead of memory for negative sampling Ji et al. (2019); He et al. (2019). Recent literature attempts to link the success of their methods with the maximization of mutual information between potential representations Oord et al. (2018); Hénaff et al. (2019). However, it is not clear whether the success of methods is determined by mutual information or by the specific form of sustained loss Tschannen et al. (2019).

**Self-Supervised Methods for FSL** CACTUs Hsu et al. (2019) first developed a two-stage strategy: constructing meta-training tasks on an unsupervised set by clustering algorithms and then running MAML or ProtoNets on the constructed tasks. Comparably, UMTRA Khodadadeh et al. (2019), AAL Antoniou & Storkey (2019), and ULDA Qin et al. (2020) proposed to construct meta-training tasks by augmenting the unsupervised data and training by the ready-made ProtoNet or MAML model. These unsupervised methods for FSL focused on allocating pseudo labels to unsupervised data. Then the existing supervised FSL models can work without modification with these pseudo labels. Three latest work, LST Li et al. (2019b), ProtoTransfer Medina et al. (2020) and CC Gidaris et al. (2019), introduce self-supervised techniques Jing & Tian (2020) to achieve superior performance on FSL tasks. LST is a semi-supervised meta-learning method that meta learns how to pick and label unsupervised data to improve FSL performance further. ProtoTransfer is state-of-the-art self-supervised FSL methods. CC considers two self-supervised methods in the present FSL work: CC-Rot, predicting the rotation incurred by an image Gidaris et al. (2018), and CC-Loc, predicting the relative location of two patches from the same image Doersch et al. (2015). CC can deal with unsupervised tasks while LTS not. In this work, We reconsidered FSL theoretically and experimentally from the perspective of self-supervised learning.

## 7 CONCLUSION

In this work, we focus on theories of self-supervised FSL. We first give two assumptions and explain why self-supervised methods are suitable for few-shot learning. Then we decompose the self-supervised loss into supervised loss and a gap bounded by the intra-class representation deviation. Experimental results on two FSL benchmarks Omniglot and *mini*ImageNet verify our theories and we propose potential ways to improve test performance.

### ACKNOWLEDGMENTS

Use unnumbered third level headings for the acknowledgments. All acknowledgments, including those to funding agencies, go at the end of the paper.

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

## A    APPENDIX

## B    PROOFS.

Assume that we have access to an episodic unsupervised task $\mathcal{T}_t$ with augumented data $\mathcal{X} = \{x^q, x^+, x_1^-, \ldots, x_{N_K-1}^-\}$. Please refer to Algorithm 1 for details in our paper. We mark their ground-truth labels by $\mathcal{C}_U = \{c^q, c^+, c_1^-, \ldots, c_{N_K-1}^-\}$, respectively. Note that $x^q$, $x^+$ are drawn from the same data distribution $\mathcal{D}_{c^+}$ (since $c^q = c^+$) while negative $x_i^-$ are drawn from the data distributions $\mathcal{D}_{c_i^-}$. Let $I = \{1, \ldots, N_K - 1\}$ be the set of indices of negative data, the unsupervised loss function is as shown in Eq. 2. where $q, k^+, k_i^-$ are the representations of data $x^q, x^+, x_i^-$, respectively.

### B.1    PROOF OF THEOREM 1.

We first leverage the convexity of $\ell$ to get a lower bound of unsupervised loss $\mathcal{L}_U$ in Eq. (16). Then we decompose the lower bound into a supervised loss $\mathcal{L}_{\text{sup}}$ plus a degenerate term in Eq. (20).

***Step 1. Convexity.*** $\ell(\boldsymbol{v}) = \log(1 + \sum_i e^{\boldsymbol{v}_i}), \forall \boldsymbol{v} \in \mathbb{R}^{N_K-1}$ is a convex function. Because, $\forall t \in \mathbb{R}, \boldsymbol{z}, \boldsymbol{v} \in \mathbb{R}^{N_K-1}$,

$$
\begin{aligned}
g(t) &= \ell(\boldsymbol{z} + t\boldsymbol{v}) = \log(1 + \sum_i e^{\boldsymbol{z}_i + t\boldsymbol{v}_i}) \\
g'(t) &= \frac{\sum_i \boldsymbol{v}_i e^{\boldsymbol{z}_i + t\boldsymbol{v}_i}}{1 + \sum_i e^{\boldsymbol{z}_i + t\boldsymbol{v}_i}} \\
g''(t) &= \frac{\left(\sum_i \boldsymbol{v}_i^2 e^{\boldsymbol{z}_i + t\boldsymbol{v}_i}\right)\left(1 + \sum_i e^{\boldsymbol{z}_i + t\boldsymbol{v}_i}\right) - \left(\sum_i \boldsymbol{v}_i e^{\boldsymbol{z}_i + t\boldsymbol{v}_i}\right)^2}{\left(1 + \sum_i e^{\boldsymbol{z}_i + t\boldsymbol{v}_i}\right)^2},
\end{aligned}
\tag{13}
$$

where $\boldsymbol{z}_i, \boldsymbol{v}_i$ are the i-th component of vector $\boldsymbol{z}, \boldsymbol{v}$. And We have,

$$
\sum_i \boldsymbol{v}_i^2 e^{\boldsymbol{z}_i + t\boldsymbol{v}_i} \geq 0,
\tag{14}
$$

and Cauchy inequality,

$$
\left(\sum_i \boldsymbol{v}_i^2 e^{\boldsymbol{z}_i + t\boldsymbol{v}_i}\right)\left(\sum_i e^{\boldsymbol{z}_i + t\boldsymbol{v}_i}\right) \geq \left(\sum_i \boldsymbol{v}_i e^{\boldsymbol{z}_i + t\boldsymbol{v}_i}\right)^2.
\tag{15}
$$

Thus, $g''(t)$ are always non-negative. $\ell(\boldsymbol{v})$ is a convex function.

***Step 2. Jensen's inequality.*** The key point in the proof is the use of Jensen's inequality since $\ell(\boldsymbol{v}) = \log(1 + \sum_i \exp(\boldsymbol{v}_i)), \forall \boldsymbol{v} \in \mathbb{R}^{N_K-1}$ is a convex function.

$$
\begin{aligned}
\mathcal{L}_U &= \mathbb{E}_{q, k^+, k_i^-} \log(1 + \sum_{i \in I} \exp(\mu q^T k_i^- - \mu q^T k^+)) \\
&= \mathbb{E}_{q, c^+, c_i^-} \mathbb{E}_{k^+ \sim c^+, k_i^- \sim c_i^-} \log(1 + \sum_{i \in I} \exp(\mu q^T k_i^- - \mu q^T k^+)) \\
&\geq \mathbb{E}_{q, c^+, c_i^-} \log(1 + \sum_{i \in I} \exp(\mathbb{E}_{k^+ \sim c^+, k_i^- \sim c_i^-} (\mu q^T k_i^- - \mu q^T k^+))) \\
&= \mathbb{E}_{q, c^+, c_i^-} \log(1 + \sum_{i \in I} \exp(\mu q^T p_{c_i^-} - \mu q^T p_{c^+})),
\end{aligned}
\tag{16}
$$

where $p_c$ is a class-wise prototype, which is the mean of all representations with the same class label $c$. That is, $p_{c_i^-}$ is the prototype of class $c_i^-$ and $p_{c^+}$ is that of class $c^+$. There are some subtle differences between our class-wise prototype and the episodic prototype in ProteNets. Our $p_c$ is a global prototype for each class. But $p_c$ in ProteNets is re-calculated by support data in each episode. Note that in the lower bound quantity of unsupervised loss, the random class labels $c_i^-$ may be true negative classes or false negative classes.

***Step 3. Decompose the lower bound.*** We devide all negative classes $c_i^-$ set into two disjoint subsets, true negative classes and false negative classes. Clearly, we divide the set of negative data indices $I$ into two unjoint subsets: true negative indices $I^- = \{i \in I | c_i^- \neq c^+\}$ and false negative indices

$I^+ = \{i \in I | c_i^- = c^+\}$. We have,

$$
\begin{aligned}
& \mathbb{E}_{q,c^+,c_i^-} \log(1 + \sum_{i \in I} \exp(\mu q^\mathrm{T} p_{c_i^-} - \mu q^\mathrm{T} p_{c+})) \\
& = \mathbb{E}_{q,c^+,c_i^-} \ell(\{\mu q^\mathrm{T} p_{c_i^-} - \mu q^\mathrm{T} p_{c+}\}_{i \in I}) \\
& = P(I^+ = \emptyset) \mathbb{E}_{q,c^+,c_i^-} \left[ \ell(\{\mu q^\mathrm{T} p_{c_i^-} - \mu q^\mathrm{T} p_{c+}\}_{i \in I}) | I^+ = \emptyset \right] \\
& + P(I^+ \neq \emptyset) \mathbb{E}_{q,c^+,c_i^-} \left[ \ell(\{\mu q^\mathrm{T} p_{c_i^-} - \mu q^\mathrm{T} p_{c+}\}_{i \in I}) | I^+ \neq \emptyset \right].
\end{aligned}
\tag{17}
$$

We define $\mathcal{C}_{\mathrm{uni}}$ as the label set after de-duplicating class labels in $\mathcal{C}_U$ (class labels including positve class $c^+$ and negative classes $c_i^-$). Since we have $\ell(\{v_i\}_{i \in I_1 \cup I_2}) := \log(1 + \sum_{i \in I_1 \cup I_2} \exp(v_i)) \geq \ell(\{v_i\}_{i \in I_1}), \forall I_1, I_2 \subseteq I$, we can decompose the above quantity to handle repeated classes.

If $I^+ = \emptyset$, then $I = I^-$, we can choose all de-duplicating negative class indices as $I_{\mathrm{uni}}$. Thus $I_{\mathrm{uni}} \subseteq I^- = I$, and $\ell(\{v_i\}_{i \in I}) \geq \ell(\{v_i\}_{i \in I_{\mathrm{uni}}})$. That is,

$$
\begin{aligned}
& \mathbb{E}_{q,c^+,c_i^-} \left[ \ell(\{\mu q^\mathrm{T} p_{c_i^-} - \mu q^\mathrm{T} p_{c+}\}_{i \in I}) | I^+ = \emptyset \right] \\
& \geq \mathbb{E}_{q,c^+,c_i^-} \left[ \ell(\{\mu q^\mathrm{T} p_{c_i^-} - \mu q^\mathrm{T} p_{c+}\}_{i \in I_{\mathrm{uni}}}) | I^+ = \emptyset \right] \\
& = \mathbb{E}_{q,c^+} \left[ \ell(\{\mu q^\mathrm{T} p_c - \mu q^\mathrm{T} p_{c+}\}_{c \in \mathcal{C}_{\mathrm{uni}} \backslash c^+}) | I^+ = \emptyset \right].
\end{aligned}
\tag{18}
$$

Observe that the last expectation in the Eq. (18) is actually the supervised loss $\mathcal{L}_{\mathrm{sup}}$ by regarding $\mathcal{C}_{\mathrm{sup}} := \mathcal{C}_{\mathrm{uni}}$. The supervised loss is based on the positive class and de-duplicating negative classes of each episodic data in our SSM. It is somehow like dynamic $N$-way 1-shot supervised loss and $N$ is the number of unique classes of sampled classes $\mathcal{C}_U$.

If $I^+ \neq \emptyset$, we can choose all false negative class indices $I^+$. All these false negative data have the same class label $c^+$. Thus, $c_i^- = c^+, p_{c_i^-} = p_{c+}, \forall i \in I^+$. Since $I^+ \subseteq I$, and $\ell(\{v_i\}_{i \in I}) \geq \ell(\{v_i\}_{i \in I^+})$. That is,

$$
\begin{aligned}
& \mathbb{E}_{q,c^+,c_i^-} \left[ \ell(\{\mu q^\mathrm{T} p_{c_i^-} - \mu q^\mathrm{T} p_{c+}\}_{i \in I}) | I^+ \neq \emptyset \right] \\
& \geq \mathbb{E}_{q,c^+,c_i^-} \left[ \ell(\{\mu q^\mathrm{T} p_{c_i^-} - \mu q^\mathrm{T} p_{c+}\}_{i \in I^+}) | I^+ \neq \emptyset \right] \\
& = \mathbb{E}_{q,c^+,c_i^-} \left[ \ell(\{0\}_{i \in I^+}) | I^+ \neq \emptyset \right] \\
& = \mathbb{E}_{q,c^+,c_i^-} \left[ \log(1 + |I^+|) | I^+ \neq \emptyset \right] \\
& = \mathbb{E}_{c^+} \left[ \log(1 + |I^+|) | I^+ \neq \emptyset \right],
\end{aligned}
\tag{19}
$$

where $|I^+|$ represents the number of elements in the set $I^+$.

From the Eq. (17), Eq. (18) and Eq. (19), we get

$$
\begin{aligned}
& \mathbb{E}_{q,c^+,c_i^-} \log(1 + \sum_{i \in I} \exp(\mu q^\mathrm{T} p_{c_i^-} - \mu q^\mathrm{T} p_{c+})) \\
& \geq P(I^+ = \emptyset) \mathcal{L}_{\mathrm{sup}} + P(I^+ \neq \emptyset) \mathbb{E}_{c^+} \left[ \log(1 + |I^+|) | I^+ \neq \emptyset \right].
\end{aligned}
\tag{20}
$$

Combining Eq. (20) with Eq. (16), we have,

$$
\mathcal{L}_U \geq P(I^+ = \emptyset) \mathcal{L}_{\mathrm{sup}} + P(I^+ \neq \emptyset) \mathbb{E}_{c^+} \left[ \log(1 + |I^+|) | I^+ \neq \emptyset \right].
\tag{21}
$$

Thus we have proved the Theorem 1 using the fact that $\gamma_0 = \frac{1}{P(I^+ = \emptyset)}$, $\delta = -\frac{P(I^+ \neq \emptyset)}{P(I^+ = \emptyset)} \mathbb{E}_{c^+} \left[ \log(1 + |I^+|) | I^+ \neq \emptyset \right]$.

## B.2 PROOF OF THEOREM 2.

First, we decompose the unsupervised loss into true negative data loss $\mathcal{L}_U^-$ and false negative data loss $\mathcal{L}_U^+$ by the property of Eq. (22), and further give an upper bound for the false negative data

loss $\mathcal{L}_U^+$ by the property of Eq. (25). Finally, we get a bound for our SSM in Eq. (29) and prove Theorem 2.

**Step 1. Inequality 1 of $\ell$.** We note that the function $\ell$ satisfies the following constant: $\ell(\{v_i\}_{i \in I_1 \cup I_2}) \leq \ell(\{v_i\}_{i \in I_1}) + \ell(\{v_i\}_{i \in I_2}), \forall I_1, I_2 \subseteq I$. Because,

$$
\begin{aligned}
\ell(\{v_i\}_{i \in I_1 \cup I_2}) &= \log(1 + \sum_{i \in I_1 \cup I_2} e^{v_i}) \\
&\leq \log(1 + \sum_{i \in I_1} e^{v_i} + \sum_{i \in I_2} e^{v_i}) \\
&\leq \log[(1 + \sum_{i \in I_1} e^{v_i})(1 + \sum_{i \in I_2} e^{v_i})] \\
&= \log(1 + \sum_{i \in I_1} e^{v_i}) + \log(1 + \sum_{i \in I_2} e^{v_i}) \\
&= \ell(\{v_i\}_{i \in I_1}) + \ell(\{v_i\}_{i \in I_2}).
\end{aligned}
\tag{22}
$$

**Step 2. Decompose the unsupervised loss.** We have already divided all negative classes into two disjoint subsets and gotten their index sets $I^-, I^+$. $I^-$ is for the true negative data while $I^+$ is for the false negative data. According to these index sets and the property in Eq. (22), we have,

$$
\begin{aligned}
\mathcal{L}_U &= \mathbb{E}_{q,k^+,k_i^-} \log(1 + \sum_{i \in I} \exp(\mu q^T k_i^- - \mu q^T k^+)) \\
&= \mathbb{E}_{q,k^+,k_i^-} \ell(\{\mu q^T k_i^- - \mu q^T k^+\}_{i \in I}) \\
&\leq \mathbb{E}_{q,k^+,k_i^-} \big[\ell(\{\mu q^T k_i^- - \mu q^T k^+\}_{i \in I^-}) \\
&\quad + \ell(\{\mu q^T k_i^- - \mu q^T k^+\}_{i \in I^+})\big] \\
&= \mathbb{E}_{q,k^+,k_i^-} \big[\ell(\{\mu q^T k_i^- - \mu q^T k^+\}_{i \in I^-})\big] \\
&\quad + \mathbb{E}_{q,k^+,k_i^-} \big[\ell(\{\mu q^T k_i^- - \mu q^T k^+\}_{i \in I^+})\big] \\
&:= \mathcal{L}_U^- + \mathcal{L}_U^+,
\end{aligned}
\tag{23}
$$

where the first expectation is $\mathcal{L}^-$ and the second is $\mathcal{L}^+$ ( imagining that the true negative data and false negative data have been separated during training).

**Step 3. Inequality 2 of $\ell$.** We define $v_{\max} \in \mathbb{R}$ as the maximum component with indices in $I_1$, that is $v_{\max} := \max\{v_i\}_{i \in I_1}$. If $v_{\max} > 0$, we have

$$
\begin{aligned}
\ell(\{v_i\}_{i \in I_1}) &= \log(1 + \sum_{i \in I_1} e^{v_i}) \\
&\leq \log(1 + |I_1| e^{v_{\max}}) \\
&= \log(1 + |I_1|) + \log(e^{v_{\max}} + \frac{1 - e^{v_{\max}}}{1 + |I_1|}) \\
&\leq \log(1 + |I_1|) + v_{\max}.
\end{aligned}
\tag{24}
$$

Otherwise, $v_i \leq v_{\max} \leq 0, \forall i \in I_1$, we have $\ell(\{v_i\}_{i \in I_1}) = \log(1 + \sum_{i \in I_1} e^{v_i}) \leq \log(1 + |I_1|)$. Thus we get the inequality,

$$
\begin{aligned}
\ell(\{v_i\}_{i \in I_1}) &\leq \log(1 + |I_1|) + \max\{v_{\max}, 0\} \\
&\leq \log(1 + |I_1|) + \sum_{i \in I_1} |v_i|.
\end{aligned}
\tag{25}
$$

**Step 4. The upper bound of $\mathcal{L}_U^+$.** Using the property for the function $\ell$ in Eq. (25), we can get an upper bound for $\mathcal{L}_U^+$,

$$
\begin{aligned}
\mathcal{L}_U^+ &= \mathbb{E}_{q,k^+,k_i^-} \big[\ell(\{\mu q^T k_i^- - \mu q^T k^+\}_{i \in I^+})\big] \\
&\leq \mathbb{E}_{q,k^+,k_i^-} \left[\log(1 + |I^+|) + \sum_{i \in I^+} \big|\mu q^T k_i^- - \mu q^T k^+\big|\right],
\end{aligned}
\tag{26}
$$

where the second term acts as the penalty for representation ablility by measuring the intra-class representaion deviation.

$$
\begin{aligned}
& \mathop{\mathbb{E}}_{q,k^+,k_i^-} \left[ \sum_{i \in I^+} \left| \mu q^{\mathrm{T}} k_i^- - \mu q^{\mathrm{T}} k^+ \right| \right] \\
&= \mathop{\mathbb{E}}_{c^+} \left[ |I^+| \mathop{\mathbb{E}}_{q,k^+,k_i^- \sim c^+, i \in I^+} \left| \mu q^{\mathrm{T}} k_i^- - \mu q^{\mathrm{T}} k^+ \right| \right] \\
&\leq \mathop{\mathbb{E}}_{c^+} \left[ |I^+| \sqrt{\mathop{\mathbb{E}}_{q,k^+,k_i^- \sim c^+, i \in I^+} \left| \mu q^{\mathrm{T}} k_i^- - \mu q^{\mathrm{T}} k^+ \right|^2} \right] \\
&\leq |\mu| \mathop{\mathbb{E}}_{c^+} \left[ |I^+| \sqrt{\mathop{\mathbb{E}}_{q,k^+,k_i^- \sim c^+, i \in I^+} \|q\|_2^2 \|k_i^- - k^+\|_2^2} \right] \\
&\overset{\|q\|_2^2 = 1}{=} |\mu| \mathop{\mathbb{E}}_{c^+} \left[ |I^+| \sqrt{\mathop{\mathbb{E}}_{k^+,k_i^- \sim c^+, i \in I^+} \|k_i^- - k^+\|_2^2} \right] \\
&= |\mu| \mathop{\mathbb{E}}_{c^+} \left[ |I^+| \sqrt{\mathop{\mathbb{E}}_{k^+,k_i^- \sim c^+, i \in I^+} \|k_i^- - p_{c^+} + p_{c^+} - k^+\|_2^2} \right] \\
&= |\mu| \mathop{\mathbb{E}}_{c^+} \left[ |I^+| \sqrt{\mathop{\mathbb{E}}_{k^+ \sim c^+} 2\|p_{c^+} - k^+\|_2^2} \right].
\end{aligned}
\tag{27}
$$

All data with indices in $I^+$ have the same label $c^+$, and the expectation in Eq. (27) show the intra-class representation deviation. Mark the deviation as $s(f_k) = |\mu| \mathop{\mathbb{E}}_{c^+} \left[ \sqrt{\mathop{\mathbb{E}}_{k^+ \sim c^+} \|p_{c^+} - k^+\|_2^2} \right]$. We have a uniform class distribution, thus $(N_K - 1)$ negative data can be drawn from any class with equal probability $1/|\mathcal{C}|$. Then $\mathbb{E}|I^+| = (N_K - 1)/|\mathcal{C}|$ for any positive class $c^+$. Thus, the right expectation in Eq. (27) can be bound by $s(f_k)$, that is,

$$
\mathop{\mathbb{E}}_{q,k^+,k_i^-} \left[ \sum_{i \in I^+} \left| \mu q^{\mathrm{T}} k_i^- - \mu q^{\mathrm{T}} k^+ \right| \right] \leq \sqrt{2} \, \mathbb{E}|I^+| s(f_k).
\tag{28}
$$

From Eq. (28), Eq. (26) and Eq. (23), we have

$$
\mathcal{L}_{\mathrm{U}} \leq \mathcal{L}_{\mathrm{U}}^- + \sqrt{2} \, \mathbb{E}|I^+| s(f_k) + \mathop{\mathbb{E}}_{q,k^+,k_i^-} \left[ \log \left( 1 + |I^+| \right) \right],
\tag{29}
$$

and we have

$$
\begin{aligned}
& \mathop{\mathbb{E}}_{q,k^+,k_i^-} \left[ \log \left( 1 + |I^+| \right) \right] \\
&= P(I^+ \neq \emptyset) \mathop{\mathbb{E}}_{q,k^+,k_i^-} \left[ \log \left( 1 + |I^+| \right) \mid I^+ \neq \emptyset \right] \\
&= P(I^+ \neq \emptyset) \mathop{\mathbb{E}}_{c^+} \left[ \log \left( 1 + |I^+| \right) \mid I^+ \neq \emptyset \right].
\end{aligned}
\tag{30}
$$

Combining Eq. (29), Eq. (30) and Theorem 1, we have proved Theorem 2 by setting $\gamma_1 = \sqrt{2} \gamma_0 (N_K - 1)/|\mathcal{C}|$.

## C  EXPERIMENTS

Self-supervised Algorithm. Note that this is not our contribution. It is a basic self-supervised algorithm from MoCo He et al. (2019). The algorithm is shown in Algorithm 1.

**Datasets** We evaluate SSM on two FSL benchmark datasets, Omniglot Lake et al. (2015) and *mini*ImageNet Vinyals et al. (2016). **Omniglot** is a character image dataset containing 1623 hand-written characters from 50 alphabets. Each character contains 20 gray images drawn by different writers. We resize raw images into size of $28 \times 28$ and rotate each character by $0°$, $90°$, $180°$ and $270°$ to form 4 different classes. The 1200 characters with rotations (4800 classes) are for training, 100 characters (400 classes) for validation and 323 characters (1292 classes) for testing. *mini***ImageNet** is a downsampled image subset of the large-scale ImageNet Russakovsky et al. (2015). It consists of 100 classes with 600 RGB images of size $84 \times 84$ per class, where 64 classes

---

**Algorithm 1** Self-Supervised Algorithm

---

**Require:** unsupervised auxiliary set $\mathcal{A}=\{\ldots, x_i, \ldots\}$, number of keys per matching task $N_K$, random augmentation function $\mathrm{Aug}(\cdot)$, momentum rate $\beta$, SGD learning rate $lr$.

1: randomly initialize the encoder $f_q$, $f_k$ parameters $\theta_q, \theta_k$, metric scaling scalar $\mu$
2: **while** not done **do**
3:      sample $N_K$ data $\{x_1, \ldots, x_{N_K}\}$ from $\mathcal{A}$
4:      randomly select $x_j$ as postive data, $1 \leq j \leq N_K$
5:      augment $x_j$ into $\mathrm{Aug}(x_j)$ and $\mathrm{Aug}'(x_j)$
6:      augment $x_i$ into $\mathrm{Aug}(x_i)$, $\forall i \in \{1, \ldots, N_K\}\backslash j$
7:      *positive key*: $x^+ \leftarrow \mathrm{Aug}(x_j)$ and *negative keys*: $\{x_1^-, \ldots, x_{N_K-1}^-\} \leftarrow \{\mathrm{Aug}(x_i)\}_{i \neq j}$, *query*: $x^q \leftarrow \mathrm{Aug}'(x_j)$,
8:      representation: $k^+ = f_k(x^+)$, $k_i^- = f_k(x_i^-)$, $\forall i \in \{1, \ldots, N_K-1\}$, and $q = f_q(x^q)$,
9:      evaluate task-specific metric loss $\mathcal{L}$ by Eq. (1)
10:     back propagation update: $(\theta_q, \mu) \leftarrow (\theta_q, \mu) - lr \cdot \nabla_{(\theta_q, \mu)} \mathcal{L}$
11:     momentum update: $\theta_M \leftarrow \beta \cdot \theta_M + (1 - \beta) \cdot \theta_q$
12: **end while**

---

are for training, 16 classes for validation and 20 classes for testing. The above splits all keep the same with those used by CACTUs Hsu et al. (2019) and UMTRA Khodadadeh et al. (2019) for comparison fairness. Certainly, the labels of data in training classes have been stripped to form the unsupervised auxiliary set $\mathcal{A}$.

**Setup** For fairness, the architecture of encoder $f_q$, $f_k$ keeps aligned with that used by CACTUs and UMTRA, as well as the supervised methods like MAML and ProtoNets. It is comprised of 4 convolutional blocks, each of which is a sequential combination of 64-channel $3\times3$ convolution, batch normalization, ReLU and $2\times2$ max-pooling. The last block is followed by a flattening and a normalization to form the feature representation, which leads to 64-/1600-dimensional representations for the images from Omniglot/*mini*ImageNet, respectively. We set $lr = 0.005, \beta = 0.999$ by monitoring validation performance. For augmentation $\mathrm{Aug}(\cdot)$, we keep consistent with UMTRA: for Omniglot by randomly zeroing pixels and randomly shifting, while for *mini*ImageNet by the ready-made Auto-Augmentation Cubuk et al. (2018) model. For *mini*ImageNet, we use the augmentation model trained on CIFAR. Different augmentation methods do matter according to AAL Antoniou & Storkey (2019), CC-Rot Gidaris et al. (2019), CC-Loc Gidaris et al. (2019). Thus we keep the same augmentation setting as a basic method though we could do better by changing augmentation settings. The SGD optimizer is used for BP update for $f_q$.

**Compared Methods** Compared methods are explicitly divided into unsupervised group, supervised group and ablation group. Specifically, the unsupervised group includes not only cutting-edge CACTUs and UMTRA, but also some alternate algorithms that can work on unsupervised $\mathcal{A}$ (see Hsu et al. (2019) for more details about them). We also compare to supervised MAML Finn et al. (2017) and ProtoNets Snell et al. (2017), which are considered as the ceiling limit of the unsupervised methods. In ablation group, to explore the effectiveness of the negative numbers $(N_K - 1)$ and the number $M$ of positive keys in SSM: (1) baseline, the algorithm in Algorithm 1. (2) $N_K/4$, replacing the best $N_K$ (2048 for Omniglot, 512 for *mini*ImageNet) with $N_K/4$. (3) $M$, replacing all positive or negative keys with the mean of three different keys from three augmentations of the same data.

**Results on Omniglot** Contrast results on Omniglot in Table 3 demonstrate that SSM completely surpasses CACTUs-MAML, CACTUs-ProtoNets, UMTRA and other alternate unsupervised methods, yielding dramatic improvements regardless of $(N,K)$ settings. Another noticeable observation is the much smaller performance gap between SSM with supervised MAML and ProtoNets. For (5,5) setting, especially, SSM baseline realizes accuracy 98.09% that is very close to accuracy 98.83% by supervised MAML, although it needs to use (4800$\times$20+5$\times$5) labeled data whereas our SSM relies on only 5$\times$5 labeled images for each (5,5) classification task.

**Results on *mini*ImageNet** Table 4 contrasts SSM to other methods on *mini*ImageNet. Compared to Omniglot, the underlying complexity and ambiguity of the real-world image objects in *mini*ImageNet cause relatively lower classification accuracy. Although SSM is not finetuned on the 50$\times$5 support data, it still reaches the second-best for (5,50) setting and beats many finetuning un-

| Algorithm | (5,1) | (5,5) | (20,1) | (20,5) |
|---|---|---|---|---|
| Training from scratch | 52.50 | 74.78 | 24.91 | 47.62 |
| BiGAN $k_{nn}$-nearest neighbors | 49.55 | 68.06 | 27.37 | 46.70 |
| BiGAN linear classifier | 48.28 | 68.72 | 27.80 | 45.82 |
| BiGAN MLP with dropout | 40.54 | 62.56 | 19.92 | 40.71 |
| BiGAN cluster matching | 43.96 | 58.62 | 21.54 | 31.06 |
| BiGAN CACTUs-MAML | 58.18 | 78.66 | 35.56 | 58.62 |
| BiGAN CACTUs-ProtoNets | 54.74 | 71.69 | 33.40 | 50.62 |
| ACAI $k_{nn}$-nearest neighbors | 57.46 | 81.16 | 39.73 | 66.38 |
| ACAI linear classifier | 61.08 | 81.82 | 43.20 | 66.33 |
| ACAI MLP with dropout | 51.95 | 77.20 | 30.65 | 58.62 |
| ACAI cluster matching | 54.94 | 71.09 | 32.19 | 45.93 |
| ACAI CACTUs-MAML | 68.84 | 87.78 | 48.09 | 76.36 |
| ACAI CACTUs-ProtoNets | 68.12 | 83.58 | 47.75 | 66.27 |
| UMTRA | 83.80 | 95.43 | *74.25 | *92.12 |
| AAL-ProtoNets | 84.66 | 89.14 | 68.79 | 74.28 |
| AAL-MAML++ | *88.40 | *97.96 | 70.21 | 88.32 |
| ProtoTransfer | 88.00 | 96.84 | 72.27 | 89.08 |
| **SSM baseline** | 92.00 | 98.09 | 79.99 | 94.13 |
| SSM ($N_K/4$) | 91.48 | 97.60 | 77.90 | 92.82 |
| SSM (M $\times$3) | •92.06 | •98.23 | •80.03 | •94.31 |
| *Supervised MAML* | *94.46* | *98.83* | *84.60* | *96.29* |
| *Supervised ProtoNets* | *98.35* | *99.58* | *95.31* | *98.81* |

Table 3: Accuracy (%) on Omniglot (averaged over 1000 $N$-way $K$-shot ($N$,$K$) test tasks. •: best, *: previous best). No-SSM results are cited from papers of CACTUs and UMTRA.

| Algorithm | (5,1) | (5,5) | (5,20) | (5,50) |
|---|---|---|---|---|
| Training from scratch | 27.59 | 38.48 | 51.53 | 59.63 |
| BiGAN $k_{nn}$-nearest neighbors | 25.56 | 31.10 | 37.31 | 43.60 |
| BiGAN linear classifier | 27.08 | 33.91 | 44.00 | 50.41 |
| BiGAN MLP with dropout | 22.91 | 29.06 | 40.06 | 48.36 |
| BiGAN cluster matching | 24.63 | 29.49 | 33.89 | 36.13 |
| BiGAN CACTUs-MAML | 36.24 | 51.28 | 61.33 | 66.91 |
| BiGAN CACTUs-ProtoNets | 36.62 | 50.16 | 59.56 | 63.27 |
| DeepCluster $k_{nn}$-nearest neighbors | 28.90 | 42.25 | 56.44 | 63.90 |
| DeepCluster linear classifier | 29.44 | 39.79 | 56.19 | 65.28 |
| DeepCluster MLP with dropout | 29.03 | 39.67 | 52.71 | 60.95 |
| DeepCluster cluster matching | 22.20 | 23.50 | 24.97 | 26.87 |
| DeepCluster CACTUs-MAML | 39.90 | 53.97 | 63.84 | 69.64 |
| DeepCluster CACTUs-ProtoNets | 39.18 | 53.36 | 61.54 | 63.55 |
| UMTRA | 39.93 | 50.73 | 61.11 | 67.15 |
| AAL-ProtoNets | 37.67 | 40.29 | - | - |
| AAL-MAML++ | 33.30 | 49.18 | - | - |
| ULDA-ProtoNets | 40.63 | 55.41 | 63.16 | 65.20 |
| ULDA-MetaOptNet | 40.71 | 54.49 | 63.58 | 67.65 |
| CC-Rot | 41.70 | 58.64 | 68.61 | 71.86 |
| CC-Loc | 37.75 | 53.02 | 61.38 | 64.15 |
| ProtoTransfer | *45.67 | *62.99 | *72.34 | *77.22 |
| **SSM baseline** | 41.78 | 59.13 | 68.87 | 71.92 |
| SSM ($N_K/4$) | 38.96 | 54.87 | 64.72 | 67.82 |
| SSM (M $\times$3) | 42.58 | 60.12 | 69.25 | 71.98 |
| *Supervised MAML* | *46.81* | *62.13* | *71.03* | *75.54* |
| *Supervised ProtoNets* | *46.56* | *62.29* | *70.05* | *72.04* |

Table 4: Accuracy (%) on *mini*ImageNet (similar to Table 3).

supervised methods, which is a convincing proof for the effectiveness of the representation space learned by SSM.

