# OpenReview forum: "A Theory of Self-Supervised Framework for Few-Shot Learning"
_ICLR.cc/2021/Conference — Reject_

### Official Review · AnonReviewer4 · 2020-10-28
**A Theory of Self-Supervised Framework for Few-Shot Learning**

**Rating:** 4
**Confidence:** 3

**Review:**

The paper establishes a relationship between self-supervised learning (SSL) and supervised few-shot learning (FSL) method and shows that when both are equivalent. The whole analysis and proof are based upon the two main assumptions: mean classifier and balanced class training data. The paper shows that if we have a too large number of classes in the SSL, then it is equivalent to the supervised learning scenario and model enjoy the same generalization ability. Always supervised loss is the upper bound by the SSL loss.

Comment:
1: The paper theoretically connects the SSL and FSL and shows when both will be equivalent. Theorem-1 shows that the supervised loss is upper bound by SSL loss by a linear relation (mostly scale+shift) when |C|-->infinity then both loss is equivalent. It seems that Theorem-1 is trivial since it is obvious that for the large class there will be very less chance of the negative pair is incorrect (i.e. false negative). If all the negative pair is correct, then it is same as we know the class label and we make the negative pair using the class information of all samples. I believe this theorem provides less useful information for a practical perspective.

2: Theorem 2 provides the underlying factor between the L_sup and L_U, and shows that L_sup loss is upper bound by the loss of the true-negative and the intraclass variance. For the small variance, we can reduce the gap between the supervised loss and SSL loss. Once a trivial solution is when |C|--> infinity. This theorem shows then when |C| is not large still we can still focus on reducing the intraclass variance and reduce the gap.

3: It is clear that if we have large number of class, we can reduce the gap between the supervised loss and self-supervised loss, but why the large batch size help in to get a practically better result? In this case, the probability of the false-negative samples is the same, and it does not depend on the batch size. Could you please explain that? It is written that "We can increase N by increasing the total negative samples N_k", is true but in the total negative samples the probability of the false-negative will be same, and it depends on the number of class only. Then how large batch size help?

4: In the N-way and M-shot, it is intuitive that when M increase the model performance will increase, but why with the increase of the N model performance will increase?

5: Omniglot dataset has 1623 classes, while in the paper it is written that "Omniglot involves up to 4800 classes" please check that.
https://github.com/brendenlake/omniglot

---

### Official Review · AnonReviewer1 · 2020-10-30
**A theoretical trial for understanding whether self-supervised learning helps solving FSL problem**

**Rating:** 2
**Confidence:** 3

**Review:**

The paper proposes to theoretically analyze whether self-supervised learning can help FSL.
Under simplified assumptions (a simple mean classifier is used; training data is balanced; and a particular form of loss is used), the main result in Theorem 1 shows that self-supervised training loss is an upper bound of the supervised metric loss function.

The idea is interesting and inspiring. However, the analysis is less satisfactory.
The main concern is that Theorem 1 and 2 are quite loose.
They only apply for the so-called  supervised metric loss function. Is it work for any fk and fq? Can you provide more strict error bound to quantify the difference? As said in the paper, "γ0, δ are constants depending on the class distribution ρ", then how to estimate γ0, δ? If they cannot be estimated, why we need this theory? How to link this theory to the success of self-supervised learning in solving FSL problem? Or can this theory be validated empirically?

I think this paper indeed proposes an interesting direction to explore. But without answering the above questions, the current version is not complete enough to be published.

===

During discussion period, I noticed import missing references of this paper as written by Nikunj Saunshi.
Besides, the authors do not respond to any of the reviewers' questions. Hence I change my score to strong rejection.

---

### Official Review · AnonReviewer2 · 2020-10-30
**A theoretical justification for why self-supervised learning (SSL) helps few-shot learning (FSL). Make connection between SSL loss and supervised learning loss.**

**Rating:** 2
**Confidence:** 4

**Review:**


*** Key idea justification ***

This work shows that contrastive loss (for self-supervised learning) is an upper bound of cross-entropy loss (for supervised learning) and leads to a conclusion that this is the underlying reason why self-supervised learning can help supervised learning in FSL. This reasoning makes little sense with little logic.

Concretely, there exist a number of to-be-answered questions before connecting the two things and making theoretical conclusion:
1) Why we need to know the upper bound of supervised learning loss given that we already have label data with the training data?
2) Decreasing SSL loss does not necessarily mean that supervised learning loss is also decreased, as it is just an upper bound. No guarantee there.
3) Assume SSL helps decrease the supervised learning loss, then why is this needed when we can simply use class labels to minimize it? Intuitively, the two are overlapping and SSL should be not useful.

Besides, this paper only considers the case of contrastive loss which involves false negative samples. What if applying other SSL loss function, for example rotation? I do see the same analysis applies to that.

In conclusion, the proposed theory makes little sense and is also over-claimed. The whole study is neither theoretical nor logical.


*** Presentation clarity ***

1) In general, the presentation of this paper is poor. One reason is using odd/strange terminologies and equation expressions. For example, contrastive loss (Eq 1) and cross-entropy loss (Eq 3) both are not given in their common expression. Other examples are "Supervised Metric for Representations" and "Self-Supervised Metric (SSM) for Representations", "a metric loss", etc.

2) Quite a few equations are hard to read and understand. First, Eq (1) and (3) are not expressed in a standard way. How are they derived?

3) What is the difference between a class-wise prototype pc and an episodic mean of support samples (At the end of Sec 3).

4) What means by "the class distribution ρ is uniform" in the proof of Theorem 2?

5) What is implied by the last sentence of Sec 4: Theoretically, if given an unsupervised set with infinite classes and data, the performance achieved by SSM can be very close to that by supervised training?



*** Grammatical errors ***
1) a episodic -> an episodic

---

### Official Review · AnonReviewer3 · 2020-10-31
**Nice motivation and some good ideas. Need to improve writing and empirical validation.**

**Rating:** 4
**Confidence:** 4

**Review:**

This paper performs theoretical analysis of the relationship between supervised learning (SL) and self-supervised learning (SSL) in the context of few-shot learning (FSL). It aims to quantify the gap in training loss between SL and contrastive SSL on FSL tasks by casting SSL as an SL problem. Using this formulation, the authors show that the self-supervised training loss is an upper bound of the supervised metric loss function, implying that if you reduce the self-supervision loss to be small enough, you can control the model’s supervision loss on the training data, and thus improve results on the downstream FSL tasks. The theoretical formulation also provides guidelines for the optimal values for the queue size in contrastive SSL, which the authors evaluate on omniglot and miniImageNet datasets, showing that the test performance varies with queue size.

Strengths: The motivation to perform theoretical analysis on the utility of SSL for few-shot learning is a good one. While I could not check the proofs thoroughly, they seem to provide a nice framework for explaining why SSL might provide good performance on few-shot learning.

Weaknesses and suggestions: 1. The paper is very difficult to follow. While the theory section (Section 4) is reasonably well-written, the rest of the paper needs a substantial rewrite to improve clarity and accessibility. Unfortunately the writing quality makes it difficult  to make a strong case for the paper. 2. The experiments only touch upon one aspect of theory discussed in the paper -- the impact of N and M on test performance. A more  thorough comparison with SL based few-shot learning and the impact of other factors like number of classes and class imbalance on test performance would make the paper stronger.

---

### Official Review · AnonReviewer5 · 2020-11-06
**Poor writing hampers an otherwise interesting study of a simple method**

**Rating:** 3
**Confidence:** 3

**Review:**

#### Summary
- The authors analyze a self-supervised learning framework for downstream (supervised) few-shot classification. The self-supervised stage is a simplified version of MoCo (He et al. 2019) and relies on class-invariant augmentation of unlabeled data to produce samples for a contrastive loss. This produces two encoder networks that are used in the subsequent few-shot learning stage via a distance-based classification scheme similar to that used by Snell et al. (2017), [1], [2], and Chen et al. (2019).

- The authors show that the method minimizes an upper bound on an oracle supervised distance-based classification loss. They then further analyze the looseness by decomposing the self-supervised loss into contributions from false-negative and true-negative samples. They relate these quantities to key methodological considerations, such as the level of diversity in the meta-training/base data and the number of negative samples to use during contrastive learning.

- The authors assess this method on the Omniglot and miniImageNet few-shot datasets, following the setup proposed by Hsu et al. (2019) in which the meta-training (aka base) split is treated as unlabeled. The results are strong, though are curiously relegated entirely to the Appendix.

#### Strengths
- The overall pipeline is to my knowledge novel, even though the authors are careful to state that the method is not a core contribution as it draws heavily from prior methods. Unlike previous works that consider unsupervised/self-supervised pre-training for few-shot learning, this work provides some theoretical justification for its method.

- Due to the judicious choice of considering contrastive learning and distance-based classification, the resulting analysis is relatively straightforward.

#### Weaknesses
- This submission is overall poorly written. It was very difficult to parse due to a copious number of grammatical errors. In numerous instances, I can't quite discern what the authors mean. Aside from this, there are many vague statements unsupported by reference or argument.

- The organization leaves much to be desired. For example, results of an ablation take center stage in the main text, while key experimental exposition and benchmark results are left entirely to the Appendix.

- Comparison to CACTUs (Hsu et al., 2019) is not entirely fair as the method (like most modern contrastive learning methods) requires the specification of instance transformations that are class-invariant for test tasks. This should be noted. (Though comparison to UMTRA (Khodadadeh et al., 2019) is fair.)

#### Recommendation
- I currently recommend rejection (3), as the submission's poor writing severely hampers clarity and thus prevents it from meeting publication standards. If the writing were fixed, I would probably rate it around a 6.

#### References
- [1] Qi et al., Low-Shot Learning with Imprinted Weights, CVPR 2018
- [2] Gidaris et al., Dynamic Few-Shot Visual Learning without Forgetting, CVPR 2018

---

### Public Comment · ~Nikunj_Saunshi1 · 2020-11-16
**Missing important citations; very similar results and analysis to prior work**

The introduction of this submission states, “Almost no one analyzes why a pre-trained embedded network with self-supervised training can provide a representation for downstream FSL tasks in theory.” and the related work section does not seem to have any citations to this effect. We would like to point out that the following works [1,2,4] have theoretically analyzed representations learned from contrastive learning on downstream tasks with few samples, while [3] analyzes the same for reconstruction based self-supervised learning. While [3,4] are quite recent, [1,2] have been online since at least 6 months before the deadline.


In particular, the results and analysis in this submission bear strong resemblance to those from our work [1]. Particularly, theorems 1 and 2 from this submission look very similar to theorems 4.1 and 4.5 (also Theorem 6.1) from [1], with the definition of $\mathcal{L}\_{U}$, $\mathcal{L}\_{U}^{-}$ and $\mathcal{L}\_{sup}$ (from the proof) being very similar to $L\_{un}$, $L_{un}^{\neq}$ and $L_{sup}$ from [1].
Furthermore, the proofs of these results are primarily based on the use of Jensen’s inequality and handling of the “false negative data” using an intra-class deviation measure $s(f)$, both of which also appear in [1], as does the use of mean classifier in the supervised learning phase.
The main difference seems to be the use of different representation functions $f_q$ and $f_k$ as opposed to the same function $f$ in [1]. This, however, is a straightforward extension since the proofs in [1] do not need the functions to be the same.


If the authors benefited from looking at our results from [1], it should be cited as such, along with a discussion about the differences from [1].


[1] Arora et al., A Theoretical Analysis of Contrastive Unsupervised Representation Learning, ICML 2019

[2] Tosh et al., Contrastive estimation reveals topic posterior information to linear models, 2020

[3] Lee et al., Predicting What You Already Know Helps: Provable Self-Supervised Learning, 2020

[4] Tosh et al., Contrastive learning, multi-view redundancy, and linear models, 2020

---

> ### Author Response · Authors · 2020-11-17
> **Missing important citations**
>
> You are right, I think we should read and discuss the differences.

---

### Decision · Program_Chairs · 2021-01-07
**Final Decision**

**Decision:**

Reject

**Comment:**

This paper proposed to theoretically explain why a pre-trained embedding network with self-supervised training (SSL) can provide representation for downstream few-shot learning (FSL) tasks. The review process finds that the paper may over-claim the results and that the results seem unsatisfactory. Both Reviewer 4 and Reviewer 5 expressed concerns regarding the writing, organizing, and grammar errors of this paper. The paper needs a substantial revision to improve clarity and accessibility. As pointed out by Nikunj Saunshi’s public comment, this paper may benefit from discussing the differences from the previous works, including [1].

[1] Arora et al., A Theoretical Analysis of Contrastive Unsupervised Representation Learning, ICML 2019